# Measuring the Risk Spillover Effect of RCEP Stock Markets: Evidence from the TVP-VAR Model and Transfer Entropy

**DOI:** 10.3390/e27010081

**Published:** 2025-01-17

**Authors:** Yijiang Zou, Qinghua Chen, Jihui Han, Mingzhong Xiao

**Affiliations:** 1School of Economics, Anyang Normal University, Anyang 455008, China; zouyj@mails.ccnu.edu.cn; 2School of Systems Science, Beijing Normal University, Beijing 100875, China; 3School of Computer Science and Technology, Zhengzhou University of Light Industry, Zhengzhou 450001, China; hanjihui@zzuli.edu.cn; 4School of Artificial Intelligence, Beijing Normal University, Beijing 100875, China; xmz@bnu.edu.cn

**Keywords:** RCEP stock markets, risk spillover effect, TVP-VAR model, transfer entropy

## Abstract

This paper selects daily stock market trading data of RCEP member countries from 3 December 2007 to 9 December 2024 and employs the Time-Varying Parameter Vector Autoregression (TVP-VAR) model and transfer entropy to measure the time-varying volatility spillover effects among the stock markets of the sampled countries. The results indicate that the signing of the RCEP has strengthened the interconnectedness of member countries’ stock markets, with an overall upward trend in volatility spillover effects, which become even more pronounced during periods of financial turbulence. Within the structure of RCEP member stock markets, China is identified as a net risk receiver, while countries like Japan and South Korea act as net risk spillover contributors. This highlights the current “fragility” of China’s stock market, making it susceptible to risk shocks from the stock markets of economically developed RCEP member countries. This analysis suggests that significant changes in bidirectional risk spillover relationships between China’s stock market and those of other RCEP members coincided with the signing and implementation of the RCEP agreement.

## 1. Introduction

With the rapid development of economic globalization and financial integration, the interconnections among global stock markets have become increasingly complex. International capital flows are influenced by these growing linkages, offering investors more diverse opportunities but also deepening risk transmission across markets. While this interconnectedness optimizes global resource allocation, it also facilitates the spread of financial volatility. Notably, risks from one market can quickly propagate to others, as evidenced by past events like the Asian financial crisis, the dot-com bubble, and the U.S. subprime mortgage crisis, which began in individual markets but rapidly affected global stock markets, causing widespread economic disruption.

The world is undergoing profound changes, with the COVID-19 pandemic intensifying the complexity of the international environment and increasing instability and uncertainty in the global economy. Tensions in international trade have escalated, global trade and investment rules are evolving, and the challenges of globalization have become more pronounced. In recent years, unilateralism has risen, U.S.–China trade tensions have intensified, and the pandemic has caused severe economic impacts across countries. Against this backdrop, the ten ASEAN countries, together with China, Japan, South Korea, Australia, and New Zealand, formally concluded the Regional Comprehensive Economic Partnership (RCEP), a multilateral free trade agreement, on 15 November 2020. The RCEP covers 30% of the world’s population, 30% of the global economic output, and 27.4% of total global trade. From an economic perspective, the signing of the RCEP facilitates regional economic integration in East and Southeast Asia, thereby driving global economic growth. As of 2 June 2023, following the agreement’s official implementation in the Philippines, all 15 member states of the RCEP have completed the ratification process. As a result, the RCEP has surpassed the European Union to become the world’s largest free trade agreement, based on the total GDP of its member countries.

With the agreement’s implementation, the capital markets of RCEP member countries have become increasingly interconnected. However, as most RCEP members are developing countries with relatively immature and less stable domestic capital markets, the overall financial risks are pronounced, making them more susceptible to external shocks. Therefore, accurately measuring the changes in financial market volatility spillovers among RCEP member countries before and after the agreement’s signing is crucial. This not only deepens our understanding of the transmission mechanisms of financial crises but also helps RCEP member countries develop relevant policies, improve the agreement framework, and better prevent the spread of global financial crises.

How to measure financial market volatility spillover effects and identify their generation process has always been a focal issue in financial risk research. In theory, Solnik (1974), Stulz (1981), and Adler and Dumas (1983) pointed out the reasons for the interconnectedness of stock markets, i.e., cross-border investors allocate global assets based on the global economic development and the fundamentals of stock markets in each country [1,2,3]. King and Wadhwani (1990) proposed the concept of market contagion, arguing that economic fundamentals cannot explain the massive U.S. stock market crash in 1987 and the subsequent global stock market plunge [4]. Further, they established a multi-factor pricing model to analyze stock market volatility spillovers by examining the impact of each factor on return correlations. The results show that the economic fundamentals of each market did not play a significant role in the financial risk factors influencing the volatility spillovers among stock markets [5].

In terms of methodology, there have been continuous innovations in the analysis methods for measuring financial market volatility spillover effects. Among them, the Copula function method and the spillover index method are widely applied. Copula functions can effectively measure the dependency between variables. By leveraging this property to calculate CoVaR, one can examine the risk spillover relationships among different financial markets [6,7,8,9]. However, although Copula functions can accurately measure nonlinear spillover relationships, they are less effective in reflecting the time-varying nature of nonlinear risk correlations and cannot provide the direction of risk spillovers between financial markets. To address these limitations, Diebold and Yilmaz (2009) constructed a spillover index based on the variance decomposition of the VAR model. The method uses a matrix form to reflect the magnitude and direction of overall spillover risks and can calculate the net spillover and net inflow effects of different markets [10].

However, the spillover effects of financial markets’ volatility evolve with the changes in the global economic situation and financial market structures. To obtain time-varying financial market volatility spillover effects, Diebold and Yilmaz further employed a rolling-window VAR approach to estimate the corresponding parameters within fixed-window samples, enabling time-varying spillover indices to be calculated [11,12,13]. This method has significant advantages, as it can estimate the time-varying total spillover effects across all financial markets in the sample range and analyze the time-varying net spillover and net inflow effects among different markets [14,15,16,17,18,19,20]. Despite its widespread application, the rolling-window method also reveals certain limitations, the most notable of which is the need to manually set the window width. An excessively long or short window can significantly impact the estimation results. To address this shortcoming, Antonakakis and Gabauer (2017) proposed a spillover index calculation method based on the Time-Varying Parameter Vector Autoregression (TVP-VAR) model [21]. This method eliminates the need for rolling windows and can accurately measure risk spillover effects at any point in time.

The TVP-VAR model has pioneered new avenues in studying financial risk spillover effects. Subsequently, many empirical studies have used this method to validate existing conclusions and gradually extend its application to other areas. For instance, Balcilar et al. introduced a new extended joint connectivity method based on the TVP-VAR model to analyze the connectivity of 11 agricultural commodities and crude oil futures prices between 1 July 2005 and 1 May 2020. The results indicate that system-wide dynamic connectivity changes over time and is driven by economic events [22]. Gabauer and Gupta employed the extended TVP-VAR connectivity method from Antonakakis and Gabauer (2017) [21] to study the spillover effects of internal and external categories of economic policy uncertainty (EPU) between the United States and Japan. Their results show that monetary policy uncertainty is the primary driver, followed by uncertainties related to fiscal, monetary market, and trade policies [23]. Adekoya and Oliyide applied the TVP-VAR method to examine the volatility spillover effects between commodities and financial assets while employing linear and nonlinear (quantile causal) causality tests to assess the impact of COVID-19 on inter-market connectivity. Their findings demonstrate that the two indicators of the COVID-19 pandemic (equity market volatility based on infectious diseases and the growth rate of reported U.S. COVID-19 cases) have significant causal effects on inter-market connections, particularly at low to mid-quantiles [24]. Khoury et al. employed the TVP-VAR and DCC-GARCH t-Copula models to analyze the impact of the Russia–Ukraine conflict on financial markets. Their results indicated that in developed economies, ESG (environmental, social, and governance) assets and the MSCI Index were net transmitters of risk, while gold and renewable energy were net recipients before and during the conflict [25]. Chen et al. used a TVP-VAR model combined with the spillover index to study the dynamic spillover effects of trade policy uncertainty (TPU) on the precious metals market during the U.S.–China trade war. The findings reveal that TPU from both China and the U.S. has significant spillover effects on the precious metals market, with the strength and direction of the spillover effects showing time-varying and asymmetric characteristics [26].

In addition to the connectedness approach based on statistical regression, recent studies have introduced entropy-based methods to analyze stock market dynamics [27,28,29,30,31]. Methods such as mutual information and transfer entropy offer several advantages, including being data-driven, model-free, and applicable in both linear and nonlinear contexts [32,33,34]. Transfer entropy, a metric rooted in information theory, measures the flow of information or causal relationships within a system [35,36]. Specifically, it quantifies the amount of information (or risk) transmitted from one market to another [37,38,39]. As a non-parametric, model-free approach, transfer entropy does not rely on specific statistical models, providing greater flexibility in handling complex data [40,41,42].

Therefore, this paper, based on the above considerations, utilizes the Time-Varying Parameter Vector Autoregression (TVP-VAR) model and transfer entropy to study the volatility spillover effects between China’s stock market and the stock markets of RCEP member countries. This research has significant academic and practical value. On one hand, it helps analyze the size and direction of risk transmission between stock markets of countries with close capital flows within the agreement, providing a reference for financial regulatory authorities to improve the agreement’s rules and framework. On the other hand, it aids in accurately understanding the external shocks faced by China’s capital market in the current complex and changing international environment, enabling better responses to and avoidance of external risks.

The remainder of this paper is organized as follows: Section 2 presents the construction method of the time-varying volatility spillover model; Section 3 introduces the data and their descriptive statistics; Section 4 discusses the empirical analysis results; and Section 5 provides the main conclusions of the paper.

## 2. Methodology

### 2.1. Time-Varying Parameter Vector Autoregression (TVP-VAR) Model

In recent years, using the spillover index method to examine the risk spillover effects between stock markets in various countries has become a research focus. The traditional decomposition spillover index model is easily affected by factors such as variable sorting. Based on this, Antonakakis and Gabauer proposed a method based on the Time-Varying Parameter Vector Autoregression (TVP-VAR) model. This method combined with the Diebold and Yilmaz (DY) method [10,11,13] can construct a new time-varying spillover index and avoid the problem of information loss caused by artificially setting rolling windows. Therefore, this article mainly draws on the TVP-VAR model constructed by Antonakakis and Gabauer to measure the volatility risk spillover effect between the stock markets of China and RCEP member countries. The construction process of this method is as follows: A TVP-VAR model with a lag length of order *p* is expressed as(1)$$y_{t}=a_{0}+A_{1}y_{t-1}+\cdots +A_{p}y_{t-p}+\varepsilon_{t},$$
where $y_{t}$ is the vector of log returns of *N* stock markets, $a_{0}$ is the intercept vector, $A_{1},\cdots ,A_{p}$ are the coefficient matrices, and $\varepsilon_{t}$ is the error vector. The components are independent and identically distributed.

Let $\beta_{t}=\mathrm{vec}\left(a_{0},A_{1},\cdots ,A_{p}\right)$ and $x_{t}=I⊗\left(1,y_{t-1},\cdots ,y_{t-p}\right)$. Let us assume that the coefficients $\beta_{t}$ follow a random walk process:(2)$$y_{t}=\beta_{t}x_{t}+\varepsilon_{t},$$(3)$$\beta_{t}=\beta_{t-1}+\delta_{t},$$
where $\delta_{t}∼N\left(0,\Omega \right)$. After determining the form of the TVP-VAR model, Monte Carlo simulation is used to estimate the time-varying parameters.

Next, based on the above model, the variance share is divided into self-variance share and cross-variance share by performing variance decomposition on the covariance matrix. This process is called generalized forecast error variance decomposition (GFEVD) and was proposed by Koop et al. (1996) [43] and Pesaran and Shin (1998) [44]. In this process, the *H* step forecast variance of $x_{i}$ is impacted by part $x_{j}$ and can be expressed as(4)$$d_{ij}\left(h\right)=\frac{\sigma_{ii}^{-1}\sum_{h=0}^{H}{\left(e_{i}^{T}A_{h}\sum e_{j}\right)}^{2}}{\sum_{h=0}^{H}{\left(e_{i}^{T}A_{h}\sum A_{h}^{T}e_{j}\right)}^{2}},$$
where Σ is the variance matrix of error vector $\varepsilon_{t}$; $\sigma_{ii}$ is the variance sequence of error vector $\varepsilon_{t}$; $e_{i}$ is an N×1 vector, where the *i* element is 1 and the rest are 0; *H* represents the prediction step; and $A_{h}$ is the coefficient of the moving average. The above variance decomposition matrix $d_{ij}\left(h\right)$ is formed by stacking the variance contributions of different markets, that is,(5)$$D_{ij}\left(h\right)=\left[\begin{array}{ccc} d_{11} & \cdots & d_{1N}\\ \vdots & \ddots & \vdots \\ d_{N1} & \cdots & d_{NN} \end{array}\right],$$

This matrix is an asymmetric matrix, whose diagonal elements represent the spillover intensity to itself, and the elements other than the diagonal elements represent the spillover intensity to other markets. Therefore, the intensity of market *k* receiving risk spillovers from other markets can be obtained by adding the elements other than the diagonal elements in the *k* row of the matrix, and the intensity of market k receiving risk spillovers from other markets can be obtained by adding the diagonal elements in the *k* column.

Next, we use $C_{k\leftarrow j}^{H}$ to represent the directional spillover effect that market k receives from other markets (From), that is,(6)$$C_{k\leftarrow j}^{H}=\sum_{j=1,j\neq k}^{N}d_{kj}.$$

Let $C_{i\to k}^{H}$ represent the directional spillover effect of market k on other markets (To), that is,(7)$$C_{i\to k}^{H}=\sum_{i=1,k\neq i}^{N}d_{ik},$$
then, by subtracting the risk spillover effect of market *k* from other markets, we can obtain the net risk spillover effect of market *k*:(8)$$C_{k}^{H}=C_{i\to k}^{H}-C_{k\leftarrow j}^{H}.$$

By summing up the risk spillover effects of all markets, we can obtain the total risk spillover effect of stock markets in various countries or regions, that is,(9)$$C^{H}=\frac{1}{N}\sum_{i,j=1}^{N}d_{ij}.$$

### 2.2. Transfer Entropy

The concept of thermodynamic entropy was introduced by Rudolf Clausius in 1850 as a physical quantity to measure the uniformity of energy distribution within a system [45]. The idea of incorporating entropy into information transmission was introduced by Shannon in 1948 with the concept of information entropy [46]. In 2000, Schreiber provided the definition of transfer entropy, which analyzes the interactions between systems and captures the direction of information flow between them [47]. Therefore, transfer entropy contains directional and dynamic information. The construction process of transfer entropy is as follows.

If there is a single random variable *X* and the corresponding probability distribution is p(x)=Prob(X=x], then the information entropy is defined as the average value of information under the whole probability distribution, which is(10)$$H\left(X\right)=-\sum_{x\in X}p\left(x\right)\log p\left(x\right),$$

When the joint probability distribution of multiple random variables (e.g., *X* and Y) is p(x,y), the joint entropy is defined as(11)$$H\left(X,Y\right)=-\sum_{x\in Y}\sum_{y\in Y}p\left(x,y\right)\log p\left(x,y\right),$$
and the conditional entropy is defined as(12)$$H\left(Y|X\right)=-\sum_{x\in Y}\sum_{y\in Y}p\left(x,y\right)\log p\left(y|x\right),$$
which represents the uncertainty of Y under the condition of given X.

Based on these basic concepts, the transfer entropy is defined as follows:(13)$$TE_{Y\to X}=-\sum_{x_{n+1},x_{n}^{k},y_{n}^{j}}p\left(x_{n+1},x_{n}^{k},y_{n}^{l}\right)\log\frac{p(x_{n+1}|x_{n}^{k},y_{n}^{l})}{p(x_{n+1},x_{n}^{k})},$$
where $x_{n}^{k}$ refers to a *k*-order delay subsequence of *X* and $y_{n}^{l}$ refers to a l-order delay subsequence of Y, respectively. $TE_{Y\to X}$ indicates the degree of reduction in the uncertainty of *X* when Y is known, which quantifies the ability of Y to predict *X*.

In the stock market, $TE_{i\to j}$ denotes the degree of reduction in the uncertainty of stock j when the logarithmic return series of stock i is known. In the specific computational process, we divide the logarithmic return series into five symbolic bins as $s_{1},s_{2},s_{3},s_{4},s_{5}$ according to the 0∼20%, 20%∼40%, 40%∼60%, 60%∼80%, and 80%∼100% fractions of the return interval, and then use these symbolic sequences to calculate TE between each pair of stocks [36,48]. Note that we set k=l=1 for simplicity.

## 3. Data and Descriptive Statistics

### 3.1. Data

This paper selects the daily closing prices of stock price indices from 10 RCEP member countries as sample data, based on the level of regional economic development and data availability. The full sample period spans from 3 December 2007 to 9 December 2024, as shown in Table 1. The total market capitalization of the 10 stock markets selected in this study accounts for more than half of the total market capitalization of RCEP member countries’ stock markets, making the sample data both broad and representative. All data are sourced from the Wind database.

### 3.2. Descriptive Statistics

This paper follows the general method of financial risk measurement and uses the logarithmic returns of stocks as sample data. The specific processing method is as follows:(14)$$R_{t}=\ln\left(y_{t}\right)-\ln\left(y_{t-1}\right),$$
where $R_{t}$ is the return on the *t*-th trading day, $y_{t}$ is the closing price on the *t*-th trading day, and $y_{t-1}$ is the closing price on the previous trading day.

Table 2 presents the descriptive statistics of stock return data. In terms of return volatility, the stock return rates of each country exhibit varying degrees of fluctuation. Among them, the standard deviation of the return rates for the MSCI Vietnam Index and Japan’s Nikkei 225 index are the largest, showing the most dramatic fluctuations, while the standard deviation of the return rate for the FTSE Malaysia Composite Index is the smallest, indicating the least volatility. In addition, the skewness of the daily return rates for the sample countries is all less than 0, exhibiting a distinctly left-skewed distribution, and the kurtosis of the returns is all greater than 3, showing significant leptokurtic (fat-tail) characteristics. According to the JB_Stat statistic, all series are non-normally distributed, further confirming that the stock markets of the sample countries exhibit a leptokurtic distribution. Moreover, after conducting the ADF test, the return rates of the stock indices from the 10 countries are stationary at the 1% significance level.

## 4. Empirical Results

### 4.1. Static Risk Spillover Analysis

The static risk spillover index refers to the average value obtained by summing the spillover indices at each time point. According to the AIC, the lag order of the TVP-VAR model is set to 4, and the forecast horizon for the generalized variance decomposition is set to 10. The static spillover indices for the stock markets of various countries during the full sample period are shown in Table 3. Since the risk spillover index matrix is asymmetric, its diagonal elements represent the intensity of the spillover risk within each stock market. Therefore, in Table 3, the row data, excluding the diagonal elements, represent the intensity of risk spillovers received by a particular stock market from other stock markets (From values), while the column data, excluding the diagonal elements, represent the intensity of risk spillovers from a particular stock market to others (To values). The Net value, obtained by subtracting the From value from the To value, is used to identify the role each stock market plays in the risk spillover system. Specifically, when the Net value is greater than 0, the stock market is a net risk transmitter, and when the Net value is less than 0, it is a net risk receiver.

From the results in Table 3, it can be observed that there are significant risk spillover effects between the stock markets of the RCEP countries. First, the intensity of the risk spillover from the South Korean stock market to the Japanese stock market (13.06%) is the highest, while the intensity of risk spillovers received from the Japanese stock market (12.24%) is also the highest. For China, among all the sample countries, the intensity of the risk spillover from the Chinese stock market to the South Korean stock market is the largest, reaching 4.91%, while the intensity of risk spillovers received from the South Korean stock market is also the largest, reaching 6.88%.

Second, from the To and From values, the South Korean stock market is the most active in the overall stock market risk spillover system, with the largest total risk spillover value (69.17%) and the largest total risk inflow value (59.67%). In terms of ranking, the intensity of risk spillovers from the South Korean stock market to other stock markets is ranked as Japan (13.06%), Australia (10.65%), Singapore (7.9%), and Malaysia (7.59%), which corresponds to the ranking of risk spillovers received from these countries’ stock markets: Japan (12.24%), Australia (10.53%), Singapore (7.39%), and Malaysia (6.5%).

Except for Vietnam, the Chinese stock market has the lowest total risk spillover value (34.2%) and total risk inflow value (42.02%). Finally, from the perspective of the Net value of risk spillovers, the stock markets of Singapore, Indonesia, South Korea, Japan, and Australia all have Net values greater than 0, playing the role of net risk transmitters in the system. On the other hand, the stock markets of China, Vietnam, Malaysia, New Zealand, and Thailand all have Net values less than 0, indicating that these markets are risk receivers in the system, more susceptible to risk shocks from other stock markets.

### 4.2. Dynamic Risk Spillover Analysis

The above static volatility spillover index does not fully reflect the time-varying characteristics of the risk spillover index, and may overlook the impact of significant events on volatility spillover effects. Therefore, it is necessary to study the time-varying characteristics of stock market risk spillovers from a dynamic perspective. From Figure 1, it can be observed that the overall risk spillover index exhibits obvious time-varying characteristics, which can be roughly divided into the following stages.

Stage 1: November 2008–November 2009. During this stage, a severe global financial crisis erupted, and before November 2009, the overall spillover index remained at its peak. Afterward, with countries gradually implementing adjustment measures, the impact of the financial crisis eased, and the overall volatility spillover index dropped significantly.

Stage 2: November 2009–May 2015. In this stage, due to frequent crisis events, the overall spillover index fluctuated between increases and decreases. For example, in early March 2010, the Greek sovereign debt crisis broke out, which led to a sharp rise in the overall spillover index, exceeding 70%. However, during the period from December 2012 to May 2015, the overall spillover index showed a significant and continuous decline compared with earlier periods.

Stage 3: June 2015–January 2019. During this phase, the overall spillover index also exhibited sharp fluctuations. First, affected by events such as the UK’s Brexit, the overall spillover index peaked around December 2016, reaching 75%. After that, from December 2016 to early 2018, the overall spillover index steadily declined, reaching a low of 32%. Starting from March 2018, with the outbreak of the Sino-U.S. trade conflict and a series of international events, the overall spillover index continued to rise.

Stage 4: February 2020–December 2024. Due to the negative impact of the COVID-19 pandemic, panic spread across global stock markets, and the overall risk spillover intensity of various stock markets surged to its highest point, reaching 83%. Afterward, as the COVID-19 vaccine was developed and became available, the panic situation gradually eased, and the global stock market risk spillover intensity began to adjust, with the overall dynamic spillover index showing a declining trend. However, since January 2022, with the outbreak of the Russia–Ukraine conflict, global stock markets again experienced panic, and the overall spillover index surged once more, continuing to fluctuate at a large scale.

### 4.3. Directional Spillover Effects of China’s Stock Market

To better understand the time-varying spillover volatility characteristics between China and the other sample countries, we calculated the directional spillover index between China and each country over the full sample period. The magnitude of this index indicates the strength of the spillover effect, while the sign (positive or negative) indicates the direction of the spillover. The results are shown in Figure 2.

Throughout the full sample period, the directional spillover values between China and other stock markets fluctuate significantly around 0, indicating the presence of bidirectional volatility spillover effects between China and other RCEP member countries. Before 2015, the volatility of the directional spillover index between China and countries such as South Korea, Australia, Singapore, Japan, Indonesia, Malaysia, and Thailand was relatively low, with the index being positive at a few points in time and negative most of the time. This suggests that during this period, China’s stock market exhibited a positive risk spillover effect on the stock markets of other RCEP member countries, while the Chinese stock market maintained a relatively stable situation.

During this period, the directional spillover index between China and countries like Vietnam and New Zealand showed more frequent fluctuations, indicating a significant bidirectional risk spillover relationship between China and these countries. In the period around the outbreak of the COVID-19 pandemic in 2019, the directional spillover index between China and other stock markets experienced significant up-and-down fluctuations. In particular, the directional spillover index between China and countries like Vietnam, Indonesia, New Zealand, and Malaysia fluctuated most frequently.

After the formal signing of the RCEP in 2022, the directional spillover index between China’s stock market and those of other countries mostly stayed below 0, indicating that the signing of the RCEP had, to some extent, altered the significant bidirectional risk spillover relationship between China’s stock market and those of other countries.

### 4.4. Risk Spillover Network of Stock Markets

To further characterize the risk spillover relationships between the sample countries, we follow the approach of Demirer et al. [49], utilizing complex network theory and the threshold method to construct a risk spillover network for the 10 RCEP member countries over the full sample period. This approach represents the risk spillover effects between stock markets in spatial dimensions. We use the median value of the spillover values (To values) in each stage as the threshold, ensuring that all network nodes are connected and that no local independent networks are formed. This allows for the filtering of risk spillover information between stock markets and the construction of an effective risk spillover network. Figure 3a shows the risk spillover network of the stock markets of the sample countries before the singing of the RCEP. The directed edges between two countries represent the mutual spillover effects, with the spillover effect values serving as the weights of each edge. The thicker and darker the arrow of the edge, the greater the spillover value between the two nodes. From Figure 3a, it can be observed that in the overall risk spillover network, Australia has the largest risk spillover to New Zealand, indicating that New Zealand receives the highest degree of risk spillover from Australia. Secondly, among the sample countries, South Korea is the major risk spillover country, with high spillover values to New Zealand, Vietnam, Thailand, and China. Vietnam and China are major risk input countries. For China, aside from receiving lower risk spillovers from Thailand and New Zealand, the ranking of risk inflows from other countries’ stock markets is as follows: South Korea, Singapore, Japan, and Australia.

After the formal implementation of the RCEP, significant changes occurred in the risk spillover relationships between the stock markets of the sample countries, as shown in Figure 3b. First, the number of risk spillover countries decreased from six countries during the full sample period to four countries. Indonesia, Singapore, and Malaysia shifted from risk spillover countries to risk-receiving countries. Second, among the main risk spillover countries, South Korea is no longer the largest risk spillover country and has been replaced by Australia. China remains a major risk-receiving country, but the intensity of risk inflows from other countries has decreased. Except for the high risk spillover value from South Korea, the risk inflows from other countries’ stock markets to China have all decreased.

### 4.5. Transfer Entropy Matrix

This paper calculates the transfer entropy between the stock market series of the 10 RCEP member countries over the full sample period to depict the transmission of risk information between different countries. The transfer entropy for the entire sample period is analyzed, and the countries are ranked based on the outflow, inflow, net outflow, and net inflow of risk information. The formulas for these categories are as follows.

Risk information flow out measures the intensity of risk information transmitted from one country to another. For country *i* to country *j*, it is calculated as(15)$$\mathrm{Flow}\;{\mathrm{Out}}_{i\to j}=TE_{ij}$$
where $TE_{ij}$ represents the transfer entropy from country *i* to country *j*, indicating the amount of risk information transmitted from *i* to *j*.

Risk information flow in measures the intensity of risk information received by a country from others. For country *j* receiving risk information from country *i*, it is calculated as(16)$$\mathrm{Flow}\;{\mathrm{In}}_{j\to i}=TE_{ji}$$
where $TE_{ji}$ represents the transfer entropy from country *j* to country *i*, indicating the amount of risk information received by *i* from *j*.

Net risk information flow out represents the difference between the risk information transmitted by a country and the risk information it receives from others. For country *i*, it is calculated as(17)$$\mathrm{Net}\;\mathrm{Flow}\;{\mathrm{Out}}_{i}=\sum_{j\neq i}TE_{ij}-\sum_{j\neq i}TE_{ji}$$
where $\sum_{j\neq i}TE_{ij}$ is the total amount of risk information transmitted from country *i* to all other countries, and $\sum_{j\neq i}TE_{ji}$ is the total amount of risk information received by country *i* from all other countries.

Net risk information flow in represents the difference between the risk information received by a country and the risk information it transmits to others. For country *i*, it is calculated as(18)$$\mathrm{Net}\;\mathrm{Flow}\;{\mathrm{In}}_{i}=\sum_{j\neq i}TE_{ji}-\sum_{j\neq i}TE_{ij}$$
where $\sum_{j\neq i}TE_{ji}$ is the total amount of risk information received by country *i* from all other countries, and $\sum_{j\neq i}TE_{ij}$ is the total amount of risk information transmitted by country *i* to all other countries.

Table 4 shows the results of the above analysis, which provide valuable insights into the flow of financial risks within the RCEP region. Firstly, countries such as Singapore and New Zealand act as significant hubs for both the outflow and inflow of risk information. Their roles in financial networks, either as transmitters or absorbers of risk, highlight the interconnectedness of regional economies and the importance of monitoring risk transmission channels. Secondly, China stands out as a critical player in absorbing risk information, which may point to its vulnerability to regional financial shocks despite its economic dominance. The high level of risk inflow into China suggests that its financial markets are highly sensitive to external shocks, underscoring the importance of robust risk management and monitoring systems to mitigate potential financial crises. Lastly, countries like Australia and South Korea have more balanced roles in both the inflow and outflow of risk information, indicating that they are both transmitters and receivers of risk signals. Their positioning in the global economy makes them key players in the flow of financial risks, which is crucial to understanding the broader regional risk dynamics.

For an intuitive comparison, we use heat maps to represent the transfer entropy between countries, as shown in Figure 4a,b, which represent the period before the singing of the RCEP and the period after the signing of the RCEP, respectively. The direction of the transfer entropy in this figure is from the vertical axis to the horizontal axis, and the numbers on the coordinate axis are the serial numbers of the RCEP countries. For ease of comparison, the colors of each heat map are adjusted to the same range.

By comparing the characteristics of stock market information transmission during the full-sample period and after the signing of the RCEP, several key differences can be observed. First, before the singing of the RCEP, China exhibited significant risk spillovers to Vietnam (2) and New Zealand (8), indicating that fluctuations in the Chinese market had a considerable impact on these countries. After the signing of the RCEP, China’s information transmission to Vietnam (2) and New Zealand (8) further intensified, which may reflect the deepening of regional economic cooperation brought about by the RCEP agreement. This is particularly evident in China’s increasing economic influence on Southeast Asian countries.

Second, before the singing of the RCEP, China received relatively little risk information from other countries, demonstrating strong autonomy and relatively low dependence. However, after the signing of the RCEP, the strength of risk information transmission received by China from economically developed countries such as Singapore (3) and South Korea (6) increased. This may indicate that financial ties between China and these countries have further deepened, especially following the implementation of the regional trade agreement.

Finally, before the singing of the RCEP, the transmission of risk information was relatively dispersed, particularly among South Pacific countries (such as Australia and New Zealand), where information flows were relatively weak. In contrast, after the signing of the RCEP, regional risk information transmission became more concentrated, with China, Singapore, and South Korea emerging as key financial hubs that played greater roles in both transmitting and receiving information. This suggests that the signing of the RCEP agreement has facilitated regional economic and financial market integration.

### 4.6. Network Construction by Transfer Entropy

We calculate TE between each pair of stocks through their symbolic sequences divided from logarithmic return series and then obtain TE matrices with a size of 10×10. Any element $a_{i,j}$ in each TE matrix denotes $TE_{i\to j}$. Meanwhile, by using the threshold method, we filter the TE matrices to obtain time-varying information flow networks, as shown in Figure 5a,b.

By comparing the RCEP stock market networks before and after the signing of the RCEP, we find the following: Firstly, the risk spillover relationship between China and Southeast Asian countries has increased. For example, the risk spillover relationship between China and countries like Vietnam, Malaysia, and Indonesia has strengthened. However, the risk spillover relationship between China and developed countries such as Japan and New Zealand has remained relatively stable, even though the RCEP has strengthened regional cooperation.

Secondly, the financial risk transmission between China and other emerging markets has increased. The risk spillover relationship between China and countries like South Korea, Thailand, and Singapore has increased. After the signing of the RCEP, China, as the largest economy in the region, has seen a significant increase in the impact of its financial market volatility on neighboring countries, especially in terms of risk transmission with Southeast Asian countries like Vietnam, Malaysia, and Indonesia.

Lastly, after the signing of the RCEP, the network connections have become denser. The risk spillover relationship between China and more countries, especially Southeast Asian nations, has strengthened. Additionally, from the perspective of network centrality, China’s centrality in the risk spillover network has increased after the RCEP was signed. In the event of a financial crisis within the region, China’s market may be more susceptible to volatility from other countries.

## 5. Conclusions and Policy Recommendations

### 5.1. Conclusions

This paper uses the spillover index method based on the Time-Varying Parameter Vector Autoregression (TVP-VAR) model and transfer entropy to construct the risk spillover index matrix of the RCEP member countries’ stock markets and analyzes the risk spillover effects between the sample countries’ stock markets. First, the roles and positions of the stock markets in the risk spillover system are analyzed from a static perspective. Second, the time-varying characteristics of the stock market risk spillover indices are characterized, and the net spillover effect of the Chinese stock market during the entire sample period, as well as the risk spillover effects between China and other countries’ stock markets, is explored in depth. Finally, risk spillover networks for the full sample period and after the formal signing of the RCEP are constructed, and a comparative analysis of the differences in stock market risk spillover effects in different periods is performed.

Our findings are as follows: Developed RCEP member states play an important role in the risk spillover system. The South Korean stock market is the most active in the overall risk spillover system, with the highest spillover intensity and the highest spillover inflow intensity. Except for Vietnam, China has the lowest risk spillover intensity and risk spillover inflow intensity. There are significant bidirectional volatility spillover effects between China and other stock markets. Before the signing of the RCEP, the Chinese stock market exhibited a positive risk spillover effect on other countries’ stock markets. After the formal signing of the RCEP in January 2022, the directional spillover indices between China’s stock market and those of other countries mostly remained below 0, indicating that the signing of the RCEP has, to some extent, altered the bidirectional risk spillover relationship between China’s stock market and those of other countries. The risk spillover network analysis shows that the signing of the RCEP changed the risk spillover structure of the sample countries. Although China remains a major risk-receiving country, the intensity of risk inflows from other countries has decreased.

### 5.2. Policy Recommendations

Based on the above conclusions, we believe that the RCEP member countries should take a holistic approach to jointly strengthening financial market supervision, proactively preventing and responding to both internal and external financial risks, and formulating policy measures to address unexpected events. This will provide a solid guarantee for the stable development of the financial markets in RCEP member countries. Additionally, as a large developing economy in the region, China should enhance its ability to prevent financial risk inflows from developed countries within the region. While developing trade, China should also implement measures to further reduce the financial risks imported from other countries.

Table 5 summarizes the changes in the stock market risk spillover network (constructed based on transfer entropy) among the RCEP countries before and after the signing of the RCEP agreement, along with corresponding policy recommendations. The results indicate significant shifts in the in-degree and out-degree of various countries, reflecting changes in their roles as risk transmitters or receivers within the regional financial network. For example, China and Vietnam demonstrated increased in-degree values after the agreement, highlighting their growing influence as risk receivers, while countries like Australia and Japan saw reduced in-degree values, indicating decreased external risk dependence. Based on these findings, targeted policies are proposed, including strengthening risk monitoring and management, optimizing domestic financial structures, enhancing cross-border capital flow regulation, and reducing reliance on external markets to ensure greater financial stability and resilience within the RCEP region.

## Figures and Tables

**Figure 1 entropy-27-00081-f001:**
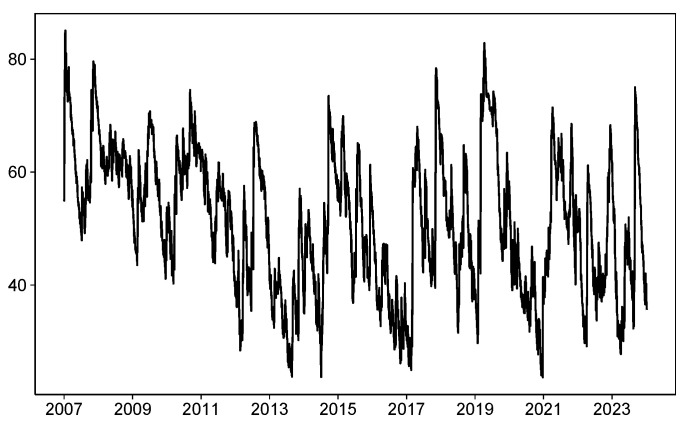
Dynamic total connectedness. Note: This figure shows the time-varying total dependency across RCEP stock markets using TVP-VAR model.

**Figure 2 entropy-27-00081-f002:**
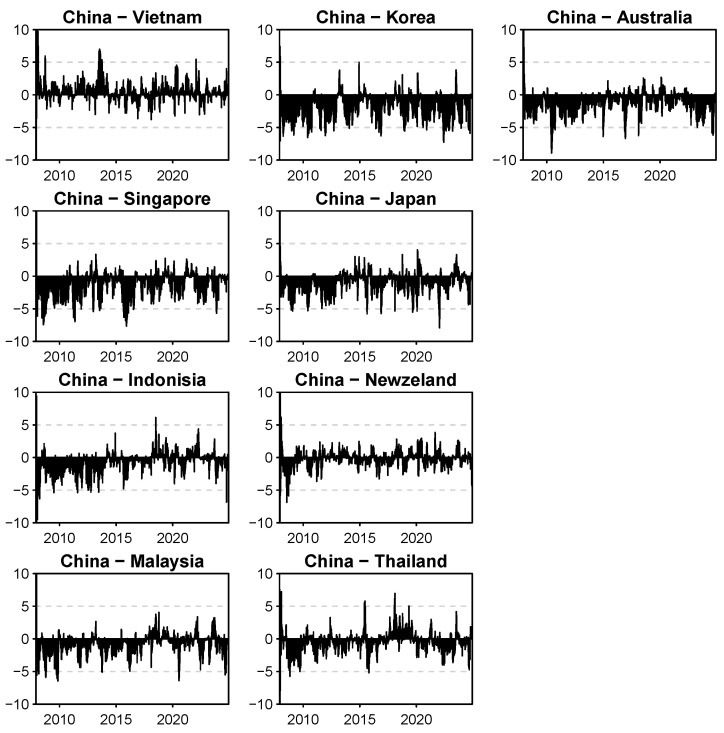
Net pairwise directional connectedness. Note: This figure only shows the directional spillover effect between China and other countries’ stock markets.

**Figure 3 entropy-27-00081-f003:**
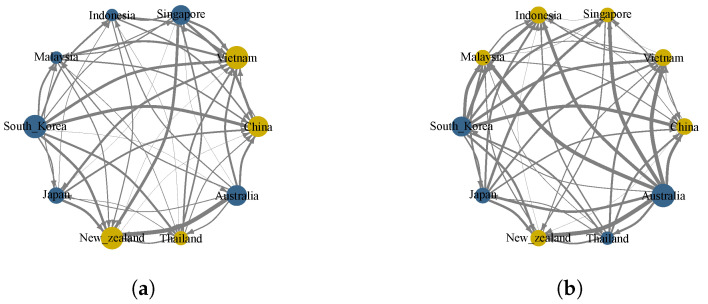
Risk spillover network of RCEP member countries’ stock markets. Note: In this figure, blue nodes represent the main risk-exporting countries, and yellow nodes represent the risk-receiving countries, and the thickness of the links represents the intensity of risk spillovers. (**a**) Before the signing of the RCEP. (**b**) After the signing of the RCEP.

**Figure 4 entropy-27-00081-f004:**
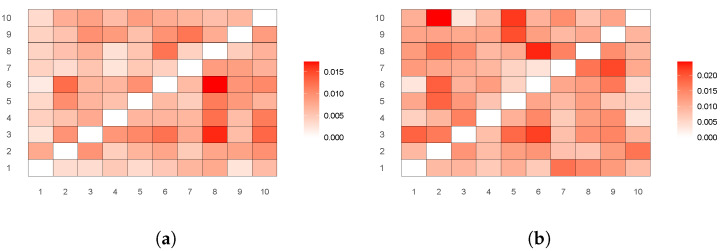
Heat maps of transfer entropy between different sectors (**a**) before the singing of the RCEP and (**b**) after the signing of the RCEP. Note: 1–10 in this figure represent the stock markets of the following 10 countries: China, Vietnam, Singapore, Indonesia, Malaysia, South Korea, Japan, New Zealand, Thailand, and Australia (arranged in order).

**Figure 5 entropy-27-00081-f005:**
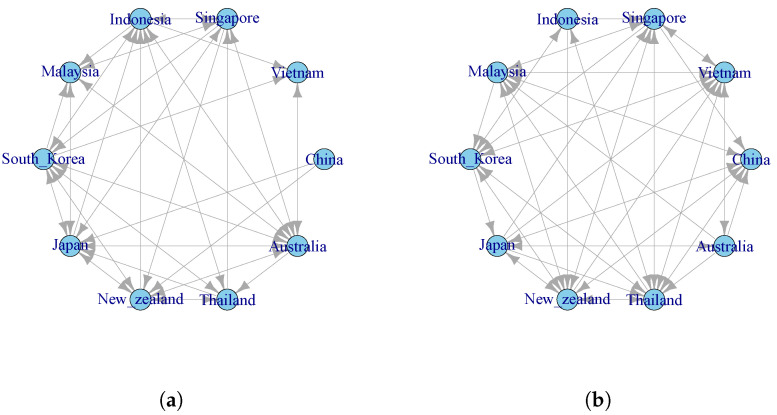
The stock market network constructed based on the transfer entropy matrix. This figure shows a directed network, and the arrows on the edges indicate the direction of information flow. (**a**) Before the singing of the RCEP. (**b**) After the signing of the RCEP.

**Table 1 entropy-27-00081-t001:** Sample country selection and stock index.

Country/Region	Stock Index	Country/Region	Stock Index
China	SSE Composite Index	South Korea	KOSPI
Vietnam	MSCI Vietnam Index	Japan	Nikkei 225
Singapore	Singapore REITS Index	New Zealand	S&P/NZX 50 Index
Indonesia	Jakarta Composite Index	Thailand	Thailand SET Index
Malaysia	FTSE Malaysia KLCI	Australia	S&P/ASX 200 Index

Note: All data are sourced from the Wind database.

**Table 2 entropy-27-00081-t002:** Descriptive statistics of stock index returns.

Country	Mean	Median	Max	Min	Std_Dev	Skewness	Kurtosis	JB_Stat	ADF_Stat
China	0.000	0.000	0.090	−0.089	0.014	−0.571	6.391	7774.574	−15.961 ***
Vietnam	0.000	0.000	0.075	−0.109	0.015	−0.423	3.214	2038.154	−15.930 ***
Singapore	0.000	0.000	0.103	−0.137	0.011	−0.171	18.889	65,815.525	−13.533 ***
Indonesia	0.000	0.000	0.097	−0.113	0.012	−0.601	11.425	24,338.786	−15.312 ***
Malaysia	0.000	0.000	0.066	−0.100	0.007	−0.746	15.654	45,601.178	−16.180 ***
South Korea	0.000	0.000	0.113	−0.112	0.012	−0.539	10.644	21,107.137	−16.604 ***
Japan	0.000	0.000	0.132	−0.132	0.015	−0.546	9.675	17,481.940	−16.358 ***
New Zealand	0.000	0.000	0.069	−0.079	0.007	−0.556	10.516	20,623.057	−15.952 ***
Thailand	0.000	0.000	0.077	−0.115	0.011	−1.260	16.344	50,431.017	−14.654 ***
Australia	0.000	0.000	0.068	−0.102	0.011	−0.692	8.371	13,278.582	−16.339 ***

Note: JB_Stat denotes the Jarque–Bera test statistic, and ADF_Stat denotes the unit root test result. “***” indicates the 1% significance level.

**Table 3 entropy-27-00081-t003:** Results of static risk spillover index.

	China	Vietnam	Singapore	Indonesia	Malaysia	South Korea	Japan	New Zealand	Thailand	Australia	From
China	57.98	2.78	6.32	4.68	4.24	6.88	4.83	2.94	4.56	4.81	42.02
Vietnam	3.44	68.14	3.92	3.12	3.79	3.73	3.92	3.02	3.49	3.44	31.86
Singapore	4.74	2.7	44.04	7.07	6.81	7.9	6.18	5.22	6.93	8.4	55.96
Indonesia	3.73	2.3	7.76	49.07	8.74	7.1	4.84	3.54	7.36	5.57	50.93
Malaysia	3.39	2.53	7.42	8.95	47.61	7.59	5.64	3.93	6.88	6.07	52.39
South Korea	4.91	2.2	7.39	6.14	6.5	40.33	12.24	3.98	5.77	10.53	59.67
Japan	3.81	2.62	6.35	4.58	5.16	13.06	43.07	4.88	4.64	11.81	56.93
New Zealand	2.83	2.56	6.66	4.03	4.45	5.26	6.1	52.89	3.96	11.25	47.11
Thailand	3.86	2.6	7.83	7.72	7.09	7	5.07	3.33	50.37	5.13	49.63
Australia	3.49	2.32	8.07	5.05	5.35	10.65	11.25	8.73	4.54	40.56	59.44
To	34.2	22.61	61.73	51.34	52.13	69.17	60.07	39.57	48.11	67.01	505.93
Net	−7.82	−9.26	5.77	0.41	−0.26	9.5	3.14	−7.54	−1.52	7.57	

**Table 4 entropy-27-00081-t004:** Risk information flow and net flow analysis for RCEP countries based on transfer entropy matrix.

Rank	Risk Information Flow Out	Risk Information Flow In	Risk Information Net Flow Out	Risk Information Net Flow In
1	Singapore	New Zealand	Singapore	New Zealand
2	South Korea	Australia	Indonesia	Australia
3	Thailand	South Korea	Thailand	Japan
4	Vietnam	Singapore	South Korea	Malaysia
5	Indonesia	Vietnam	China	Vietnam
6	Malaysia	Malaysia	Vietnam	China
7	Australia	Thailand	Malaysia	South Korea
8	New Zealand	Japan	Japan	Thailand
9	Japan	Indonesia	Australia	Indonesia
10	China	China	New Zealand	Singapore

**Table 5 entropy-27-00081-t005:** Policy recommendations after signing of RCEP.

Country	Before Signing of RCEP		After Signing of RCEP		Policy Recommendations
	**In_Degree**	**Out_Degree**	**In_Degree**	**Out_Degree**	
China	0	2	6	4	Strengthen the monitoring of external financial risks, especially the financial links with other RCEP member countries.
Vietnam	3	0	7	5	Strengthen the control of inbound financial risks, especially the regulation of capital inflows.
Singapore	5	5	6	6	Continue to leverage its advantages as a regional financial center, strengthen the regulation of financial markets, and ensure market transparency and stability.
Indonesia	5	4	2	4	Focus on changes in the transmission of risks within the region and reduce the spillover effects of potential financial shocks.
Malaysia	5	2	6	5	Focus on enhancing the risk management capabilities of domestic financial institutions to reduce the impact of external financial volatility.
South Korea	7	7	6	5	Further optimize the structure of the capital market to reduce the impact of regional financial volatility on the domestic market.
Japan	8	7	4	6	Reduce dependence on external market fluctuations and enhance the independence and stability of the domestic market.
New Zealand	5	4	7	7	Strengthen the management of cross-border capital flows to avoid excessive dependence on external capital.
Thailand	5	4	8	6	Strengthen the early warning mechanism for external financial risks to reduce the impact of external risks on the domestic market.
Australia	7	8	2	4	Shift the policy focus to the domestic financial market to reduce sensitivity to regional financial risks.

## Data Availability

The raw data supporting the conclusions of this article will be made available by the authors on request.

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
