# Peer review of "Measuring the Risk Spillover Effect of RCEP Stock Markets: Evidence from the TVP-VAR Model and Transfer Entropy"

_entropy, 2025, doi:10.3390/e27010081_

Round 1
Reviewer 1 Report
Comments and Suggestions for Authors
The research presented in this paper applies existing methods to study risk spillovers amongst RCEP countries. I find this inquiry interesting and relevant. The manuscript makes a very nice contribution to the special issue.
My criticisms are quite limited as I found the analysis to be sound.
Abstract. The authors have a tendency to draw far-reaching conclusions which are not really supported by the findings. To give just one example, I cite from the abstract: "This highlights the current "fragility" of China’s stock market, making it susceptible to risk shocks from the stock markets of economically developed RCEP member countries." The current downside is largely home-made with a property crises of previously unknown proportions. While I believe the findings on risk-spillovers to hold water, drawing conclusions which ignore ideosyncratic shocks can be misleading.
I am a bit puzzled about the second paragraph of the introduction which is a bit over the top.
"The world today is undergoing profound changes unseen in a century. ...."
In the last century there was a second world war, oil crises a dot-com bubble, a financial crash in 2007/8 etc.
It would be preferable to make this intro less sensational and more down-to-earth.
"As a result, RCEP has surpassed the European Union to
become the world’s largest free trade agreement." I guess you mean by total GPD of its member countries.
"Figure 2." this is really hard to see. Changing the scale of the y-axis should help.
"Figure 5." I find this figure hard to read (I cannot really see the arrows) an also hard to interpret as the position of countries moved in addition to the vertices.
Author Response
Comment 1: The research presented in this paper applies existing methods to study risk spillovers amongst RCEP countries. I find this inquiry interesting and relevant. The manuscript makes a very nice contribution to the special issue.My criticisms are quite limited as I found the analysis to be sound.
Response: Thanks a lot for the positive comments.
---------------------------------------------------------------------------------------------------
Comment 2: Abstract. The authors have a tendency to draw far-reaching conclusions which are not really supported by the findings. To give just one example, I cite from the abstract: "This highlights the current "fragility" of China’s stock market, making it susceptible to risk shocks from the stock markets of economically developed RCEP member countries." The current downside is largely home-made with a property crises of previously unknown proportions. While I believe the findings on risk-spillovers to hold water, drawing conclusions which ignore ideosyncratic shocks can be misleading.
Response: Regarding the concerns raised about the abstract, we have made revisions based on the reviewer’s feedback, specifically by removing the inappropriate conclusions mentioned. We recognize that the discussion on the "fragility" of China’s stock market might have been too broad and did not sufficiently consider the multiple internal and external factors affecting China’s market. Therefore, we have removed the related statements from the abstract to avoid any misleading inferences.
---------------------------------------------------------------------------------------------------
Comment 3: I am a bit puzzled about the second paragraph of the introduction which is a bit over the top.
"The world today is undergoing profound changes unseen in a century. ...."
In the last century there was a second world war, oil crises a dot-com bubble, a financial crash in 2007/8 etc.
It would be preferable to make this intro less sensational and more down-to-earth.
Response: Thank you for your thoughtful consideration. To make the language less sensational and more grounded, We have revised the original statement as suggested. The modified version now presents the context in a more balanced and objective manner while maintaining the core message.
For convenience, the revised sentence has been highlighted in red in the manuscript (the second paragraph of the introduction).
---------------------------------------------------------------------------------------------------
Comment 4: "As a result, RCEP has surpassed the European Union to
become the world’s largest free trade agreement." I guess you mean by total GPD of its member countries.
Response: We have revised the original statement in the manuscript to clarify that RCEP becoming the world’s largest free trade agreement is based on the total GDP of its member countries.The revised content has been highlighted in red in the manuscript (the second paragraph of the introduction).
---------------------------------------------------------------------------------------------------
Comment 5: "Figure 2." this is really hard to see. Changing the scale of the y-axis should help.
Response: We have adjusted the scale of the y-axis to improve the readability of Figure 2. The updated figure is highlighted in red in the manuscript at the relevant reference.
---------------------------------------------------------------------------------------------------
Comment 6: "Figure 5." I find this figure hard to read (I cannot really see the arrows) an also hard to interpret as the position of countries moved in addition to the vertices.
Response: We have adjusted the network layout by adopting a circular layout to ensure that the positions of nodes remain consistent before and after the signing of RCEP, facilitating easier comparison. Additionally, we increased the size of the edge arrows to make the direction of the arrows more visible. Finally, the RCEP spillover network, constructed using the transfer entropy matrix, is intended primarily as a visualization tool to observe the direction of risk spillovers, and it has certain limitations in terms of understanding the network's structural characteristics.The updated figure is highlighted in red in the manuscript at the relevant reference.
---------------------------------------------------------------------------------------------------
Reviewer 2 Report
Comments and Suggestions for Authors
The purpose of the present article is to study the volatility spillover effects between China's stock market and the stock markets of RCEP member countries. In order to accomplish this objective, research methods such as the time-varying parameter vector autoregressive (TVP-VAR) model and transfer entropy were utilized. Furthermore, emphasis was placed on the theoretical and practical significance of the research. On the one hand, the necessity of ascertaining the magnitude and direction of risk transmission between capital markets of countries bound by economic agreements was underscored. On the other hand, an effort was made to elucidate external shocks occurring in the Chinese capital market.
The article demonstrates a satisfactory degree of scientific rigor. The methodology employed is appropriate, the text's structure is logical, and the hypotheses have been formulated correctly, though they lack explicit articulation. However, there are several imperfections in the text that must be addressed. These are as follows:
1. It appears that the authors have failed to differentiate between thermodynamic entropy and information entropy. It is imperative to acknowledge that Clausius is credited with formulating the definition of thermodynamic entropy, while Shannon is recognized for his contributions to the definition of information entropy (lines 183-186).
2. As illustrated in Figure 5, the clarity of the figure is suboptimal, and the directionality of the information flow is not readily discernible due to the blurring of the arrows.
3. A summary of the study's findings for each of the RCEP countries is imperative. The most effective method for accomplishing this is through tabular presentation. The table should encompass the period preceding and following the ratification of the economic agreement. In addition to the interdependencies that have been previously examined, the table should also include the financial policies that have been recommended for each nation. The implementation of this approach will yield three primary benefits: enhanced article quality, heightened reader engagement, and augmented citability.
Author Response
Comment 1: The purpose of the present article is to study the volatility spillover effects between China's stock market and the stock markets of RCEP member countries. In order to accomplish this objective, research methods such as the time-varying parameter vector autoregressive (TVP-VAR) model and transfer entropy were utilized. Furthermore, emphasis was placed on the theoretical and practical significance of the research. On the one hand, the necessity of ascertaining the magnitude and direction of risk transmission between capital markets of countries bound by economic agreements was underscored. On the other hand, an effort was made to elucidate external shocks occurring in the Chinese capital market.
Response: Thanks a lot for the positive comments.
---------------------------------------------------------------------------------------------------
Comment 2: The article demonstrates a satisfactory degree of scientific rigor. The methodology employed is appropriate, the text's structure is logical, and the hypotheses have been formulated correctly, though they lack explicit articulation. However, there are several imperfections in the text that must be addressed. These are as follows:
It appears that the authors have failed to differentiate between thermodynamic entropy and information entropy. It is imperative to acknowledge that Clausius is credited with formulating the definition of thermodynamic entropy, while Shannon is recognized for his contributions to the definition of information entropy (lines 183-186).
Response: Thank you very much for your valuable reminder. We have made the distinction between the thermodynamic entropy proposed by Clausius and the information entropy introduced by Shannon in the manuscript. The relevant sentences have been revised, with the changes highlighted in red.
---------------------------------------------------------------------------------------------------
Comment 3: As illustrated in Figure 5, the clarity of the figure is suboptimal, and the directionality of the information flow is not readily discernible due to the blurring of the arrows.
Response: We have adjusted the network layout by adopting a circular layout to ensure that the positions of nodes remain consistent before and after the signing of RCEP, facilitating easier comparison. Additionally, we increased the size of the edge arrows to make the direction of the arrows more visible. The updated figure is highlighted in red in the manuscript at the relevant reference.
---------------------------------------------------------------------------------------------------
Comment 4: A summary of the study's findings for each of the RCEP countries is imperative. The most effective method for accomplishing this is through tabular presentation. The table should encompass the period preceding and following the ratification of the economic agreement. In addition to the interdependencies that have been previously examined, the table should also include the financial policies that have been recommended for each nation. The implementation of this approach will yield three primary benefits: enhanced article quality, heightened reader engagement, and augmented citability.
Response: Thank you for your valuable suggestion. Your input has been instrumental in improving the structure of the article. In the final chapter, we have added a policy recommendation section, where we provide specific financial policy suggestions for each country based on the changes in the risk spillover network structure before and after the RCEP agreement. The relevant content is presented in Table 5. The revised sections are highlighted in red for your reference.
---------------------------------------------------------------------------------------------------
Reviewer 3 Report
Comments and Suggestions for Authors
The manuscript investigates the risk spillover dynamics among stock markets of RCEP member countries, which is a relevant and timely topic, considering the growing interest in financial market spillovers and regional economic integration. It uses a Time-Varying Parameter Vector Autoregression (TVP-VAR) model and transfer entropy methods, which are suitable and contemporary econometric techniques. The paper is well-structured, with a logical progression from methodological description to empirical analysis and discussion.
My main concern is that the paper seems to attribute the changes in spillover and entropy between the periods before and after November 15, 2020, directly to the signing and implementation of RCEP. However, such an attribution implies causality, which the employed methods do not fully support. The methods can capture patterns and changes in financial interconnectedness but cannot explain the reasons behind these changes. External factors such as the global economic environment, monetary policies, trade tensions, or the COVID-19 pandemic could have influenced the observed dynamics.
Ideally, a parallel analysis involving non-RCEP countries or other regional groups as a control could help disentangle the impact of RCEP from broader global economic trends. However, I recognize that such an analysis would require substantial additional work and data collection, which may be beyond the scope of the paper. Therefore, I am not requesting this addition but suggest that the authors acknowledge this limitation explicitly in the paper. In particular, some of the language should be toned down and revised to avoid implying causality. For example, the sentence in the abstract, "The signing and implementation of the RCEP have, to some extent, altered the bidirectional risk spillover relationships between China’s stock market and those of other RCEP members." could be rephrased to: "The analysis suggests that significant changes in bidirectional risk spillover relationships between China’s stock market and those of other RCEP members coincided with the signing and implementation of the RCEP agreement."
Additionally, I am wondering how the authors have considered that the exchanges from the ten countries may not have the same trading days, for example, due to local holidays. Such inconsistencies could introduce noise in the calculation of spillover indices and transfer entropy. Could the authors please clarify how they handled this issue? For example, were missing data interpolated, or were non-trading days synchronized across countries?
Author Response
Comment 1: The manuscript investigates the risk spillover dynamics among stock markets of RCEP member countries, which is a relevant and timely topic, considering the growing interest in financial market spillovers and regional economic integration. It uses a Time-Varying Parameter Vector Autoregression (TVP-VAR) model and transfer entropy methods, which are suitable and contemporary econometric techniques. The paper is well-structured, with a logical progression from methodological description to empirical analysis and discussion.
Response: Thanks a lot for the positive comments.
---------------------------------------------------------------------------------------------------
Comment 2: My main concern is that the paper seems to attribute the changes in spillover and entropy between the periods before and after November 15, 2020, directly to the signing and implementation of RCEP. However, such an attribution implies causality, which the employed methods do not fully support. The methods can capture patterns and changes in financial interconnectedness but cannot explain the reasons behind these changes. External factors such as the global economic environment, monetary policies, trade tensions, or the COVID-19 pandemic could have influenced the observed dynamics.
Ideally, a parallel analysis involving non-RCEP countries or other regional groups as a control could help disentangle the impact of RCEP from broader global economic trends. However, I recognize that such an analysis would require substantial additional work and data collection, which may be beyond the scope of the paper. Therefore, I am not requesting this addition but suggest that the authors acknowledge this limitation explicitly in the paper. In particular, some of the language should be toned down and revised to avoid implying causality. For example, the sentence in the abstract, "The signing and implementation of the RCEP have, to some extent, altered the bidirectional risk spillover relationships between China’s stock market and those of other RCEP members." could be rephrased to: "The analysis suggests that significant changes in bidirectional risk spillover relationships between China’s stock market and those of other RCEP members coincided with the signing and implementation of the RCEP agreement."
Response: Thank you very much for your valuable comments. We believe your suggestions are very important, particularly regarding the attribution of causality and the potential influence of external factors on the observed changes in spillover effects. Due to limitations in space and data collection, this paper indeed lacks a parallel analysis involving non-RCEP countries or other regional groups. We will continue to focus on this aspect in our future work.
We also agree that the modification to the abstract you proposed is both necessary and appropriate. As such, we have made the suggested revision to the abstract, with the changes highlighted in red.
---------------------------------------------------------------------------------------------------
Comment 3: Additionally, I am wondering how the authors have considered that the exchanges from the ten countries may not have the same trading days, for example, due to local holidays. Such inconsistencies could introduce noise in the calculation of spillover indices and transfer entropy. Could the authors please clarify how they handled this issue? For example, were missing data interpolated, or were non-trading days synchronized across countries?
Response: Thank you for raising this important issue. We acknowledge that the original data collected indeed contains inconsistencies in trading days across countries, which could introduce noise into the calculation of spillover indices and transfer entropy. To address this, we have cleaned and pre-processed the data, specifically by removing the dates with inconsistent trading days, ensuring that the trading data for all countries remain synchronized.
---------------------------------------------------------------------------------------------------
Round 2
Reviewer 3 Report
Comments and Suggestions for Authors
I am pleased with the authors' answers and their revised manuscript.